# Association between Vitamin D Deficiency and Clinical Parameters in Men and Women Aged 50 Years or Older: A Cross-Sectional Cohort Study

**DOI:** 10.3390/nu15133043

**Published:** 2023-07-05

**Authors:** Ji Hyun Lee, Ye An Kim, Young Sik Kim, Young Lee, Je Hyun Seo

**Affiliations:** 1Department of Internal Medicine, Veterans Health Service Medical Center, Seoul 05368, Republic of Korea; jhleemd626@gmail.com (J.H.L.); yeanin@gmail.com (Y.A.K.); kys1217@bohun.or.kr (Y.S.K.); 2Veterans Medical Research Institute, Veterans Health Service Medical Center, Seoul 05368, Republic of Korea; lyou7688@bohun.or.kr

**Keywords:** vitamin D deficiency, glycated hemoglobin, diabetes mellitus, body mass index, obesity, tumor necrosis factor-alpha, inflammation

## Abstract

Vitamin D deficiency (VDD) is increasingly prevalent on a global scale and is connected to chronic health issues including diabetes, obesity, and inflammation. This study aimed to investigate the association between VDD and various clinical parameters including glycated hemoglobin (HbA1c), body mass index (BMI), and inflammatory markers. This cross-sectional cohort study included Korean men and women aged 50 years and older (290 men, 125 women); VDD was classified as serum 25-hydroxyvitamin D (25[OH]D) levels below 20 ng/mL. Vitamin D deficiency was more prevalent in men (64.5%) compared to that in women (35.2%). Men with VDD had higher fat mass and HbA1c levels, lower muscle strength, and worse physical performance. Among women, VDD was associated with higher BMI, HbA1c, tumor necrosis factor-alpha (TNF-α), and creatinine levels. In women, 25(OH)D levels exhibited an inverse relationship with HbA1c, BMI, and TNF-α concentrations. However, there were no differences in the levels of interleukin-6 and interleukin-1 beta according to vitamin D status in both men and women. Vitamin D deficiency is linked to higher HbA1c, BMI, and inflammatory markers in older Korean women, thus warranting the maintenance of sufficient vitamin D levels for overall health.

## 1. Introduction

Vitamin D deficiency (VDD) is increasingly prevalent on a global scale, predominantly attributed to increasingly inactive indoor routines and a tendency to avoid sunlight. The Institute of Medicine (IOM) has established 25-hydroxyvitamin D (25[OH]D) levels as a biomarker with proven validity in evaluating vitamin D status. Values below 20 ng/mL are regarded as inadequate, while values 20 ng/mL or higher are deemed sufficient [1]. While the impact of vitamin D on bone health is well-accepted, its associations with non-skeletal chronic health issues have gained recognition [2]. Recent evidence has also indicated a connection between vitamin D and metabolic diseases, including diabetes. An inverse relationship has been noted between 25(OH)D levels and glucose [3,4]. Aly YE et al. demonstrated the negative relationships between levels of 25(OH)D and glucose, insulin, and Homeostasis Model Assessment (HOMA) [5]. There is a notable association between VDD and diabetes, as both conditions are influenced by similar factors, including obesity, low physical activity, and aging [6].

Obesity, marked by the excessive buildup of body fat, is a complex condition. Vitamin D is a fat-soluble nutrient present within various body tissues including fat, muscle, the liver, and serum [7]. Interestingly, a higher body mass index (BMI) has been linked to lower baseline levels of 25(OH)D [8]. The association between obesity and VDD suggests an interplay between these factors. It is possible that the increased fat in individuals with obesity sequesters vitamin D, leading to lower circulating levels [9,10].

Moreover, vitamin D has been found to possess potent immune-modulatory activities [11,12], and decrease the concentration of inflammatory markers including tumor necrosis factor-alpha (TNF-α) and interleukin-6 (IL-6) [13] while up-regulating the production of interleukin-1 beta (IL-1β) [14]. In Hashimoto’s thyroiditis, a positive correlation was observed between vitamin D levels and TNF-α, IL-5, and IL-17 [15]. Vitamin D3 supplementation in diabetes mellitus and autoimmune thyroiditis reduced the concentration of inflammatory type 1 T helper cytokines (INF-γ, TNF-α, and IL-6), and increased levels of anti-inflammatory type 2 T helper cytokines (IL-4, IL-5) [16]. However, the results of these studies were inconsistent. Vitamin D replacement has been beneficial for reducing C-reactive protein in patients with diabetes but has not shown a significant impact on TNF-α and IL-6 levels [17]. Similarly, no association was observed between vitamin D levels and serum TNF-α in individuals with thyroiditis and Graves’ disease [18].

Therefore, this cross-sectional cohort study aimed to investigate the association between vitamin D levels and clinical parameters, including glycated hemoglobin (HbA1c), BMI, and inflammatory markers in Korean men and women aged ≥50 years.

## 2. Materials and Methods

### 2.1. Participants and Study Design

The participants for this study were selected from the Veterans Health Service Medical Center (VHSMC) sarcopenia cohort, which consisted of individuals with and without sarcopenia who visit the Endocrinology Department of the VHSMC in Seoul, Korea, between July 2020 and December 2020 (IRB No. 2020-02-015) [19]. The study design received approval from the Institutional Review Board (IRB No. 2022-06-010).

Data from the VHSMC sarcopenia cohort and the VHSMC Information System (ezCaretech, Korea) were used as eligibility criteria. The data included pharmacy records, laboratory data, and disease diagnoses. We excluded participants with a previous history of stroke, acute myocardial infarction, angina, active malignancy, liver disease, and dementia with functional decline or limitation. All participants completed surveys that covered various aspects such as smoking habits, alcohol consumption, exercise, vitamin D supplementation, the EuroQol Visual Analog Scale (EQ-VAS), and the strength, assistance with walking, rising from a chair, climbing stairs, and falls (SARC-F) questionnaire.

Among the VHSMC Sarcopenia cohort, a total of 321 men and 144 women participated. However, after excluding individuals without blood samples (*n* = 44) and those aged below 50 years (*n* = 6), the final analysis included 290 men and 125 women (Figure 1). The participants’ vitamin D status was categorized based on the IOM guidelines, which classified individuals with 25(OH)D levels below 20 ng/mL as having VDD, while those with 25(OH)D levels ≥20 ng/mL were considered non-deficient in vitamin D [1]. The height measurements were recorded in centimeters (cm), whereas weight measurements were recorded in kilograms (kg) without shoes and with light clothes.

Body mass index (BMI) was calculated using the following formula: weight (kg) divided by height squared (m^2^). Fat and lean mass were evaluated and adjusted using an In-body (720) body composition analyzer (Biospace Co., Seoul, Korea).

### 2.2. Laboratory Tests

All participants underwent an 8-h fasting period prior to the collection of blood samples from the antecubital vein. The same machine in a single laboratory was used to analyze various indicators in the collected samples. These indicators included fasting blood glucose, HbA1c, creatinine, and 25(OH)D levels. Serum 25(OH)D concentrations were measured using the chemiluminescent microparticle immunoassay (CMIA) using an Architect i2000SR system (Abbott, Singapore). The TNF-α, IL-6, and IL-1β concentrations in the serum were quantified using the Human Luminex^®^ Discovery assay kit (R&D Systems, Inc., Minneapolis, MN, USA) as per the manufacturer’s instructions [20]. The estimated glomerular filtration rate (eGFR) of the kidneys was calculated using the Chronic Kidney Disease Epidemiology Collaboration (CKD-EPI) equation [21].

### 2.3. Covariates Assessment

The smoking status of participants was categorized into three categories: Never smoked, former smoker, and current smoker. Former smokers were individuals who had consumed at least five packs of cigarettes or had quit smoking within 6 months prior to the baseline assessment. Alcohol intake was identified as alcohol consumption at least once per week. Exercise was identified as engaging in physical activity that caused the body to perspire, performed at least three times per week, and categorized as moderate-intensity exercise. Participants were identified as having diabetes mellitus if their HbA1c levels were equal to or higher than 6.5%, or if their fasting plasma glucose levels after an 8-h fasting period were 126 mg/dL or higher while concurrently taking anti-diabetic medications, including insulin, at the time of the survey.

### 2.4. Assessment of Sarcopenia

Body composition was measured via bioelectrical impedance analysis using an InBody 570 device (Biospace Co., Seoul, Korea). The calculation of appendicular skeletal muscle mass (ASM) involved adding the muscle masses of both the arms and legs. To determine the skeletal muscle mass index (SMI), ASM was adjusted relative to the height squared. Muscle strength was evaluated using handgrip strength (HGS), which was measured using a digital hand dynamometer (T.K.K 5401, Takei, Tokyo, Japan). During the measurement, the participants stood with their arms lowered and feet positioned shoulder-width apart. They gripped the bar with their second finger and exerted maximum force on the dynamometer using both hands. Handgrip strength was assessed twice on each side, and the average values were used. The 5-times chair stand test (CST) measured the time taken to perform five repetitions of standing up and sitting down, from an armchair with a straight back, without utilizing the arms. Participants completed the test with their arms folded across their chests [22]. In this study, low muscle mass was categorized as an SMI below 7.0 kg/m^2^ for men and below 5.7 kg/m^2^ for women. Low muscle strength was identified as HGS below 28 kg in men and below 18 kg in women. Low physical performance was identified as having a CST time equal to or greater than 12 s or failing the pre-test according to the consensus of the Asian Working Group for Sarcopenia in 2019 [23].

### 2.5. Statistical Analysis

Statistical analyses were performed using SPSS 22.0 (IBM, Inc., Armonk, NY, USA). The choice between parametric (Student’s *t*-test) or non-parametric (Mann–Whitney U test) methods was based on the normality test results for continuous variables. Continuous variables were reported as the mean ± standard deviation (SD) or median [interquartile range], while categorical data were analyzed utilizing the chi-square test and expressed as numbers (percentages).

Linear correlation analysis was applied to examine the associations between 25(OH)D and clinical parameters including BMI, HbA1c, and TNF-α. A correlation figure was created using GraphPad Prism 9.3.0 (GraphPad software, La Jolla, CA, USA). In order to verify whether an association persists even after controlling for confounding factors, we conducted an additional analysis using multiple linear regression. Model 1 was adjusted for age. Model 2 was adjusted for age, smoking, alcohol drinking, and exercise. In the final model (Model 3), we adjusted for confounding factors used in Model 2 plus eGFR, diabetes, hypertension, and vitamin D replacement. The results were reported as the beta (B), standard error (SE), 95% confidence interval, and *p*-value. A *p*-value less than 0.05 (*p* < 0.05) was considered significant.

## 3. Results

### 3.1. Clinical Characteristics of the Study Participants

The study participants’ clinical characteristics are summarized in Table 1. Among 290 men, 187 (64.5%) were classified as having VDD (25[OH]D < 20 ng/mL), while 44 (35.2%) of the women were classified as VDD. A notable disparity was observed between men and women, indicating a higher prevalence of VDD among men than in women. The mean serum 25(OH)D levels were 18.1 ± 10.6 ng/mL in men and 25.1 ± 12.1 ng/mL in women.

Table 2 presents the characteristics of the study participants based on vitamin D status. Men with VDD had significantly higher mean fat mass and HbA1c levels than those without VDD. Men with VDD also had lower HGS and performed worse in CST than men without VDD. No significant differences were observed in age, BMI, exercise, alcohol consumption, smoking history, lean mass, inflammatory markers, or sarcopenia among the different vitamin D status groups. Among women, those with VDD had significantly higher BMI, HbA1c, TNF-α, and creatinine levels than those without VDD. However, no significant differences were observed in mean IL-6, IL-1β, HGS, and CST values between women with and without VDD. Women without VDD were more likely to exercise three or more times per week than those with VDD (75.3% vs. 56.8%). Vitamin D replacement was more commonly observed in the group without VDD than in the group with VDD, in both men (49.5% vs. 24.6%) and women (60.5% vs. 22.7%).

### 3.2. Association between Vitamin D and Clinical Parameters

The correlation analysis revealed significant inverse relationships between 25(OH)D levels and HbA1c, BMI, and TNF-α concentrations in women (Figure 2). In women, after adjusting for confounding factors, the multiple linear analysis consistently showed negative associations between HbA1c, BMI, TNF-α, and vitamin D in all three models (Figure 3, Appendix A Appendix A). However, no significant associations were observed between these variables in men. Similarly, no significant correlations were found between 25(OH)D levels and age, IL-6, IL-1β, HGS, and CST in both men and women (Appendix A Appendix A).

## 4. Discussion

Within the scope of this cross-sectional cohort study, a negative association was observed between serum 25(OH)D levels and HbA1c, BMI, and serum TNF-α concentrations in women. This association remained significant after correcting for potential confounding factors including age, smoking, alcohol consumption, exercise, eGFR, diabetes, hypertension, and vitamin D replacement.

Furthermore, individuals with VDD, both men and women, exhibited higher mean HbA1c levels compared to those without VDD. This finding is reinforced by a randomized controlled trial demonstrating the beneficial impacts of vitamin D on insulin sensitivity and diabetes risk in individuals with prediabetes and hypovitaminosis D [24]. The United States (US) National Health and Nutrition Examination Survey revealed a negative association between 25(OH)D and HbA1c levels in participants without diabetes mellitus [25]. Several mechanisms have been proposed to explain the connection between vitamin D levels and glucose metabolism. Vitamin D receptors (VDRs) present in the pancreas, muscles, and adipocytes can influence insulin secretion, glucose uptake, and insulin resistance [26]. The activation of VDRs within pancreatic β cells directly influences the secretion of insulin [27], while in skeletal muscle, it enhances glucose uptake through the SIRT1/IRS1/GLUT4 axis [28]. Furthermore, vitamin D has been found to impede adipocyte differentiation and adipogenesis and decrease peroxisome proliferator-activated receptor-γ expression [29]. These actions contribute to the reduction in peripheral insulin resistance, which can be beneficial for individuals with obesity and disorders of metabolism.

The inverse association between 25(OH)D and BMI was consistent with the findings of a previous meta-analysis [30]. Women with VDD tended to exhibit higher BMI than those without VDD. Similarly, in men, our study found that fat mass was higher in those with VDD than in those without, indicating a potential association between vitamin D status and body composition. These findings align with those of a study by Araghi et al., who documented a significant correlation between higher BMI, higher fat mass, and lower serum 25(OH)D levels [31]. The fat tissue is the major storage site for 25(OH)D [9]. Consequently, it has been proposed that VDD may occur due to the reduced availability of vitamin D resulting from its accumulation in the fat tissue and its subsequent gradual release into the bloodstream [9,10]. Obesity is strongly linked to low-grade inflammation, which is primarily attributed to the excessive production of pro-inflammatory markers by adipose tissue and adipocytes, along with concurrent VDD [32]. The link between VDD and insulin resistance may be explained by inflammatory processes, as individuals with VDD have shown elevated levels of inflammatory markers [33]. In human adipocytes, vitamin D lowers pro-inflammatory cytokines expression, including TNF-α and IL-6 [32].

Our study demonstrated an inverse correlation between 25(OH)D and TNF-α, which is consistent with prior research, indicating a negative connection between serum TNF-α concentrations and 25(OH)D levels in healthy women [34]. In patients with diabetic foot infections, vitamin D levels < 25 nmol/L were associated with elevated TNF-α concentrations [35]. Recent studies have also suggested links between TNF-α, IL-6, and COVID-19 severity, and vitamin D has been recognized for its potential as a therapeutic agent for COVID-19 owing to its immuno-modulatory effects [13]. Vitamin D possesses the capacity to suppress immune responses regulated by Th1 cells through the inhibition of pro-inflammatory cytokine production. Additionally, it induces anti-inflammatory pathways involving Th2 and T reg cells [36]. Significantly, the replacement of calcitriol for a duration of 6 months resulted in a noteworthy reduction in TNF-α levels in women with osteoporosis [37]. These results indicate a potential role of vitamin D in modulating inflammation. Vitamin D may regulate the function of immune and inflammatory cells, thereby stimulating anti-inflammatory pathways while suppressing the activation of these cells [38]. Vitamin D exerts its anti-inflammatory effects by inhibiting the NF-kB transcription factor, which is involved in the generation of pro-inflammatory cytokines [39]. TNF-α is produced by macrophages, lymphoid cells, adipocytes, and fibroblasts [40]; many of these cells also express VDRs [41,42,43]. IL-6, synthesized by various cell types including immune, vessel, and bone cells, exhibits pleiotropic properties with both pro-inflammatory and anti-inflammatory effects [44]. Vitamin D has been shown in several clinical trials to modulate IL-6 levels, potentially preventing oxidative stress and tissue damage in tuberculosis [36]. However, Kotsiou et al. found that there was no relationship between IL-6 and vitamin D levels in patients with obstructive sleep apnea [45]. Similarly, in our investigation, no relationship between IL-6 and vitamin D was observed.

Our study found that men with VDD had lower muscle strength and worse physical performance than those without VDD. These findings align with those of a prospective study conducted on participants aged ≥65 years from the Chianti region in Italy, which revealed a significant link between low vitamin D levels and diminished physical performance, as evaluated through the HGS and CST [46]. However, it is worth noting that the available literature on this topic remains contentious. For instance, cross-sectional research on elderly women did not establish a relationship between vitamin D levels and HGS [47].

Furthermore, our study indicated that women without VDD were more likely to engage in regular exercise three times per week. Previous studies have also indicated a higher likelihood of individuals with regular exercise habits, having higher vitamin D levels [48,49].

Our study conducted in South Korea revealed that among individuals aged ≥50 years, the prevalence of VDD was 64.5% in men and 35.2% in women. Vitamin D deficiency varies globally, with prevalence rates reported at 23.3% in the US [50], 34.2% in Africa [51], 40.4% in Europe [52], 53.6% in Japan [53], and 63.2% in China [54]. When comparing adults aged ≥65 years in China and the US, a higher proportion of older Chinese individuals were classified as vitamin D deficient (70.3%) compared to older American individuals (17.4%) [55]. Kim et al. conducted research to investigate the genetic variants that contribute to the susceptibility of Koreans to VDD [56]. It is important to consider the genetic predisposition to VDD in East Asians compared to that in Caucasians, even with similar sun exposure due to their shared latitude [57]. Racial differences should be taken into account and considered when comparing these findings to those of other studies.

We analyzed data from 290 men and 125 women and examined the relationship between VDD and various clinical parameters. The large cohort allowed for meaningful conclusions. Correlation analysis revealed associations between vitamin D levels and HbA1c, BMI, and TNF-α concentrations in women. Multiple linear regression analysis was adjusted for confounding factors to assess independent associations. Sex differences were observed, with varying prevalence and associations between VDD and clinical parameters in men and women.

This research had some limitations. First, its cross-sectional design only allowed the identification of associations, making it impossible to determine causality. A longitudinal study would be more informative in investigating the temporal relationship between VDD and the clinical parameters. Second, the selection of participants from a specific cohort may have introduced selection bias, thereby reducing the representativeness of the sample. This may limit the applicability of our findings to the general population. Third, the research lacked information on important confounding factors, such as nutritional intake, sun exposure, and outdoor activities, which could have influenced the observed associations.

## 5. Conclusions

Overall, this study highlighted the association between VDD and several clinical parameters. These results suggest that VDD is associated with higher HbA1c levels, BMI, and TNF-α concentrations in women; thereby emphasizing the significance of maintaining sufficient vitamin D levels, particularly in women, to improve these clinical parameters. Additional research is warranted to investigate the underlying mechanisms and potential therapeutic implications of these findings.

## Figures and Tables

**Figure 1 nutrients-15-03043-f001:**
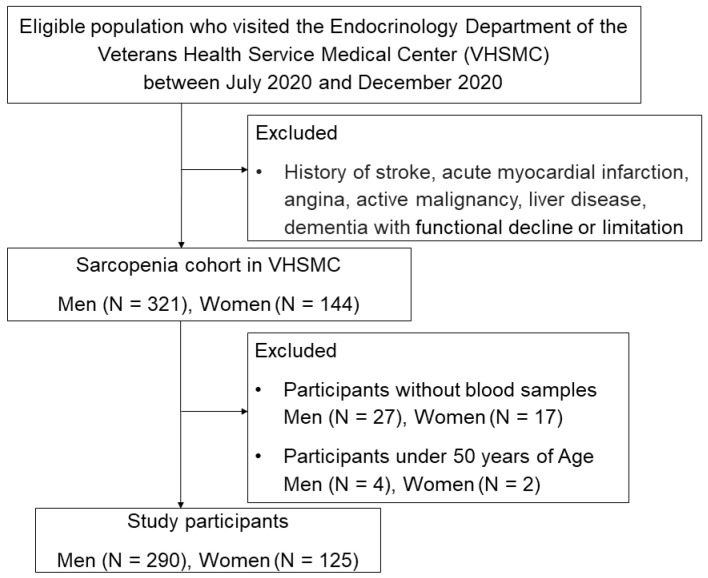
Study population.

**Figure 2 nutrients-15-03043-f002:**
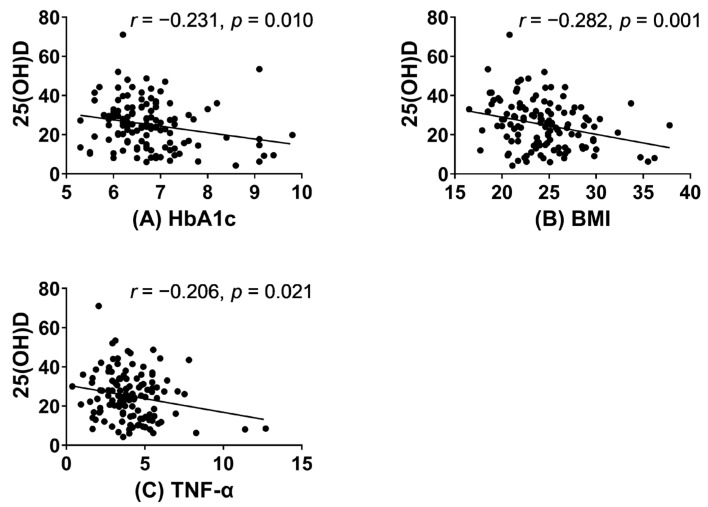
Correlations between 25(OH)D levels and HbA1c (**A**), BMI (**B**), and TNF-α (**C**) in women.

**Figure 3 nutrients-15-03043-f003:**
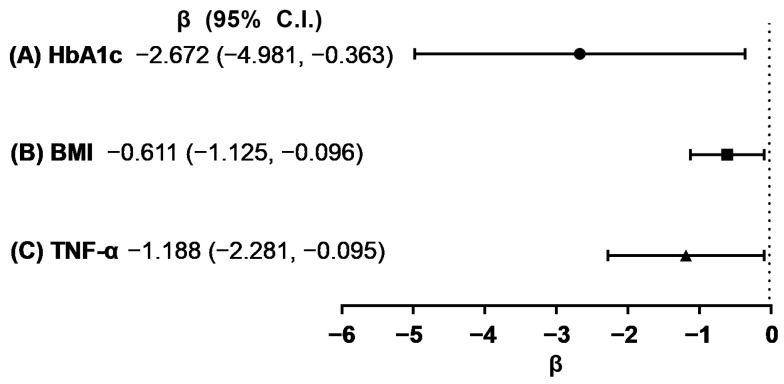
Beta-coefficient (β) and 95% confidence intervals (C.I.) from multiple linear regression with HbA1c (**A**), BMI (**B**), TNF-α (**C**), and vitamin D as outcomes in women.

**Table 1 nutrients-15-03043-t001:** Clinical characteristics of men and women.

Variables	Men (*n* = 290)	Women (*n* = 125)
Age (years)	72.7 ± 6.1	70.1 ± 5.9
Weight (kg)	65.2 ± 9.3	58.1 ± 9.7
Height (cm)	166.1 ± 5.6	154.3 ± 4.8
BMI (kg/m^2^)	23.6 ± 2.9	24.5 ± 3.9
25(OH) D (ng/mL)	18.1 ± 10.6	25.1 ± 12.1
25(OH) D < 20, *n* (%)	187 (64.5)	44 (35.2)
Exercise, *n* (%)	197 (67.9%)	86 (68.8)
Alcohol drinking, *n* (%)	145 (50.0%)	19 (15.2)
Current smoker, *n* (%)	52 (17.9%)	2 (1.6)
Vitamin D replacement, *n* (%)	97 (33.4%)	59 (47.2)
Diabetes mellitus, *n* (%)	283 (97.6)	107 (85.6)
Hand grip strength (kg)	29.4 ± 6.6	19.9 ± 4.4
Chair stand test (s)	5.0 ± 2.6	4.9 ± 3.2
Fat mass (kg)	17.3 ± 5.6	21.5 ± 6.6
Percent fat mass (%)	25.9 ± 5.8	35.7 ± 5.8
Lean mass (kg)	45.6 ± 5.3	35.4 ± 4.2
ASM (kg)	20.2 ±2.8	14.7 ±2.3
SMI (kg/m^2^)	7.3 [6.8; 7.8]	6.0 [5.6; 6.7]
Sarcopenia, n (%)	79 (27.2)	45 (36.0)
SARC-F	0.0 [0.0; 1.0]	1.0 [0.0; 2.0]
EQ-VAS	70.0 [60.0; 80.0]	70.0 [60.0; 80.0]
HbA1c (NGSP) (%)	6.9 ± 0.9	6.8 ± 0.9
TNF-α (pg/mL)	4.7 ± 2.0	4.1 ± 1.8
IL-6 (pg/mL)	2.1 ± 3.2	1.5 ± 1.6
IL-1β (pg/mL)	4.7 ± 2.0	5.7 ± 1.4
Creatinine (mg/dL)	1.0 ± 0.5	0.8 ± 0.2
CKD-EPI eGFR (*mL*/*min*/1.73 m^2^)	87.4 ± 6.9	103.4 ± 18.7

Data are expressed as the mean ± standard deviation or the median [Interquartile range], or *n* (%). Abbreviations: BMI, body mass index; ASM, appendicular skeletal muscle mass; SMI, skeletal muscle mass index; EQ-VAS, EuroQol Visual Analog Scale; HbA1c, glycated hemoglobin; CKD-EPI eGFR, Chronic Kidney Disease-Epidemiology Collaboration estimated glomerular filtration rate; SARC-F, strength, assistance with walking, rising from a chair, climbing stairs, and falls.

**Table 2 nutrients-15-03043-t002:** Clinical parameters by vitamin D status in men and women.

	Men			Women		
	25(OH)D ≥ 20 (*n* = 103)	25(OH)D < 20 (*n* = 187)	*p*-Value	25(OH)D ≥ 20 (*n* = 81)	25(OH)D < 20 (*n* = 44)	*p*-Value
Age (years)	72.7 ± 6.3	72.7 ± 5.9	ns	69.5 ± 5.7	71.2 ± 6.0	ns
Weight (kg)	64.1 ± 8.5	65.8 ± 9.6	ns	57.0 ± 9.3	60.1 ± 10.4	ns
Height (cm)	166.0 ± 5.4	166.1± 5.7	ns	154.6 ± 5.0	153.7 ± 4.6	ns
BMI (kg/m^2^)	23.2 ± 2.8	23.8 ± 2.9	ns	23.9 ± 3.8	25.4 ± 4.0	*
25(OH)D (ng/mL)	30.0 ± 8.5	11.6 ± 3.7	–	32.0 ± 9.1	12.4 ± 4.1	–
Exercise (≥3/wk), *n*(%)	69 (67.0%)	128 68.4%)	ns	61 (75.3)	25 (56.8)	*
Alcohol drinking, *n* (%)	55 (53.4%)	90 (48.1%)	ns	16 (19.8)	3 (6.8)	ns
Current smoker, *n* (%)	15 (14.6%)	37 (19.8%)	ns	0 (0)	2 (4.5)	ns
Vitamin D replacement, *n* (%)	51 (49.5%)	46 (24.6%)	***	49 (60.5)	10 (22.7)	***
Diabetes mellitus, *n* (%)	100 (97.1%)	183 (97.9%)	ns	67 (82.7)	40 (90.9)	ns
Hypertension, *n* (%)	64 (62.1)	112 (59.9)	ns	41 (50.6)	25 (56.8)	ns
Hand grip strength (kg)	30.4 ± 6.1	28.8 ± 6.9	*	20.3 ± 4.0	19.2 ± 4.9	ns
Chair stand test (s)	5.5 ± 2.4	4.8 ± 2.6	*	5.1 ± 3.1	4.4 ± 3.4	ns
Fat mass (kg)	16.2 ± 5.4	17.9 ± 5.6	*	20.8 ± 6.2	22.8 ± 7.2	ns
Percent fat mass (%)	24.8 ± 5.8	26.6 ± 5.7	*	35.1 ± 5.7	36.7 ± 5.8	ns
Lean mass (kg)	45.6 ± 5.0	45.6 ± 5.4	ns	35.0 ± 4.1	36.1 ± 4.3	ns
ASM (kg)	20.2 ± 2.6	20.3 ± 2.9	ns	14.5 ± 2.3	15.0 ±2.4	ns
SMI (kg/m^2^)	7.3 [6.9; 7.8]	7.3 [6.8; 7.8]	ns	6.0 [5.5; 6.7]	6.3 [5.7; 6.9]	ns
Sarcopenia, n (%)	32 (31.1%)	47 (25.1%)	ns	34 (42.0)	11 (25.0)	ns
SARC-F	0.0 [0.0; 1.0]	0.0 [0.0; 1.0]	ns	1.0 [0.0; 2.0]	1.0 [0.0; 2.0]	ns
EQ-VAS	70 [60; 80]	70 [55; 80]	ns	70 [60; 80]	70 [50; 80]	ns
HbA1c (NGSP) (%)	6.8 ± 0.8	7.0 ± 1.0	*	6.6 ± 0.6	7.1 ± 1.1	***
TNF-α (pg/mL)	4.6 ± 2.2	4.7 ± 2.0	ns	3.8 ± 1.5	4.6 ± 2.2	*
IL-6 (pg/mL)	2.7 ± 5.0	1.7 ± 1.2	ns	5.7 ± 1.3	5.6 ±1.5	ns
IL-1β (pg/mL)	5.4 ± 1.6	5.3 ± 1.6	ns	1.5 ± 1.9	1.5 ± 0.9	ns
Creatinine (mg/dl)	1.0 ± 0.3	1.0 ± 0.5	ns	0.7 ± 0.1	0.8 ± 0.3	*
CKD-EPI eGFR (*mL*/*min*/1.73 m^2^)	87.2 ± 6.8	87.6 ± 7.0	ns	107.0 ± 14.6	96.8 ± 23.2	*

Data are expressed as the mean ± standard deviation or the median [Interquartile range], or *n* (%). Abbreviations: BMI, body mass index; ASM, appendicular skeletal muscle mass; SMI, skeletal muscle mass index; EQ-VAS, EuroQol Visual Analog Scale; HbA1c, glycated hemoglobin; CKD-EPI eGFR, Chronic Kidney Disease-Epidemiology Collaboration estimated glomerular filtration rate; SARC-F, strength, assistance with walking, rising from a chair, climbing stairs, and falls. Statistically significant: * *p* < 0.05, *** *p* < 0.001, ns—not statistically significant.

## Data Availability

The data are not publicly available due to Institutional Review Board regulations but can be obtained upon a reasonable request to the corresponding author (jazmin2@naver.com).

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
