# Peer review of "Association between Vitamin D Deficiency and Clinical Parameters in Men and Women Aged 50 Years or Older: A Cross-Sectional Cohort Study"

_nutrients, 2023, doi:10.3390/nu15133043_

Round 1

Reviewer 1 Report

The authors studied the association between vitamin D level and clinical parameters in men and women aged 50 years or older in Korea. They assessed the vitamin D levels using serum 25-hydroxyvitamin D (25[OH]D) levels. They found that vitamin D deficiency was associated with higher HbA1c levels and lower hand grip strength in men. In women, serum 25(OH)D level negatively correlated with HbA1c, BMI, and TNF-α concentrations. This is a nice report with useful information about vitamin D deficiency and clinical parameters. The following minor issues can be resolved to improve the presentation of the current manuscript.

Minor comments:

1.     Line 66-67: “Similarly, Ke et al. (2017) not found an association with vitamin D” Please rewrite this sentence to make it clear. The current sentence has grammatical error.

2.      Line 210-233: Both section 3.2 and section 3.3 show the negative correlation between serum 25(OH)D and HbA1c, BMI, and TNF-α concentrations in women. This seems to be redundant. Please merge the two sections or explain the need of performing both linear and multivariate linear analysis. What are the differences between these two statistical analysis methods?

3.     Section 3.3: If multivariate linear analysis is necessary, please provide the analysis results for other clinical parameters as in supplementary table 1.

English is good.

Reviewer 2 Report

Dear authors;

The manuscript aimed to investigate the association between vitamin D levels and clinical parameters, including HbA1c, BMI, as well as inflammatory markers in Korean men and women aged 50 years and older.

However, before publication, some questions and suggestions must be made:

1. Plagiarism has been found. Please, rewrite sentences to avoid it.

2. Figures and Tables must be improved following Journal guidelines.

3. If the study participants were: 290 men and 125 women. Why did you declare 415 participants?

4. Although tables are well-known and commonly accepted for publication. Some figures could be added to improve the presentation of the results.

Kind regards,

Dear authors;

Professional English editing must be made before publication.

Kind regards,

Round 2

Reviewer 2 Report

Dear authors;

The manuscript aimed to investigate the association between vitamin D levels and clinical parameters, including HbA1c, BMI, as well as inflammatory markers in Korean men and women aged 50 years and older.

I would like to thankful the modifications made to improve the manuscript. 

Kind regards,

Dear Editor;

No more modifications are required.

Kind regards,